# LFA-1/ICAM-1 Adhesion Pathway Mediates the Homeostatic Migration of Lymphocytes from Peripheral Tissues into Lymph Nodes through Lymphatic Vessels

**DOI:** 10.3390/biom13081194

**Published:** 2023-07-31

**Authors:** Jia Guo, Zeyu Xu, Rachel C. Gunderson, Baohui Xu, Sara A. Michie

**Affiliations:** 1Department of Pathology, Stanford University School of Medicine, Stanford, CA 94305, USA; guojia4090@sxmu.edu.cn (J.G.); xu2zy@ucmail.uc.edu (Z.X.); rcook@gmail.com (R.C.G.); 2Department of Surgery, Stanford University School of Medicine, Stanford, CA 94305, USA; 3Center for Hypertension Care, Shanxi Medical University First Hospital, Taiyuan 030012, China; 4Department of Medicine, College of Medicine, University of Cincinnati, Cincinnati, OH 45219, USA

**Keywords:** leukocyte function-associated antigen-1, intercellular adhesion molecule-1, lymphocytes, migration, lymphatics

## Abstract

Lymphocyte function-associated antigen-1 (LFA-1) and its endothelial ligand intercellular adhesion molecule-1 (ICAM-1) are important for the migration of lymphocytes from blood vessels into lymph nodes. However, it is largely unknown whether these molecules mediate the homeostatic migration of lymphocytes from peripheral tissues into lymph nodes through lymphatic vessels. In this study, we find that, in naive mice, ICAM-1 is expressed on the sinus endothelia of lymph nodes, but not on the lymphatic vessels of peripheral tissues. In in vivo lymphocyte migration assays, memory CD4^+^ T cells migrated to lymph nodes from peripheral tissues much more efficiently than from blood vessels, as compared to naive CD4^+^ T cells. Moreover, ICAM-1 deficiency in host mice significantly inhibited the migration of adoptively transferred wild-type donor lymphocytes from peripheral tissues, but not from blood vessels, into lymph nodes. The migration of LFA-1-deficient donor lymphocytes from peripheral tissues into the lymph nodes of wild-type host mice was also significantly reduced as compared to wild-type donor lymphocytes. Furthermore, the number of memory T cells in lymph nodes was significantly reduced in the absence of ICAM-1 or LFA-1. Thus, our study extends the functions of the LFA-1/ICAM-1 adhesion pathway, indicating its novel role in controlling the homeostatic migration of lymphocytes from peripheral tissues into lymph nodes and maintaining memory T cellularity in lymph nodes.

## 1. Introduction

Lymphocytes migrate from blood vessels into lymph nodes (LNs) and peripheral tissues for immune surveillance [1,2,3,4]. Naive T cells migrate efficiently through blood vessel high endothelial venules (HEVs) into LNs, where antigen-specific memory/effector T cells are generated. These T cells exit LNs, enter the bloodstream, and migrate into peripheral tissues to execute immunologic functions. The migration of lymphocytes from blood vessels into LNs and peripheral tissues is mediated in part by adhesion molecules and chemokine receptors on lymphocytes and their cognate ligands on blood vessel endothelia [1,4,5,6].

Lymphocytes can also migrate from peripheral tissues into LNs through lymphatic vessels. Early studies have demonstrated that most CD4^+^ T cells in afferent lymph vessels were memory phenotypes [7,8,9]. Consistent with these studies, memory/effector T cells preferentially migrate into and accumulate in normal and inflamed nonlymphoid peripheral tissues [8,10,11]. Some naive T cells can also migrate from blood vessels into peripheral tissues in naive animals [12,13]. If these T cells do not proliferate or die in situ, they may eventually exit peripheral tissues and re-enter LNs through the afferent lymphatic vessels. Under homeostatic conditions, approximately 5–10% of lymphocytes in most LNs enter through the afferent lymphatic vessels [14]; this migration may contribute to the homeostatic regulation of specific subsets of lymphocytes in LNs.

Well-defined multistep cascades, which involve adhesion molecules (L-selectin and integrins) and chemokine receptors on lymphocytes and their ligands (endothelial adhesion molecules and chemokines) on HEVs, help control the migration of lymphocytes from blood vessels into secondary lymphoid tissues, such as LNs, intestinal Peyer’s patches, and bronchus-associated lymphoid tissue [15,16,17,18,19,20,21,22]. For example, in peripheral LNs (PLNs), the binding of L-selectin on circulating lymphocytes to peripheral node addressin on endothelia initiates the adhesion and rolling of the lymphocytes on HEVs. C-C motif chemokine ligand 21 (CCL21) on the luminal surface of the HEVs binds to C-C motif chemokine receptor 7 (CCR7) on the rolling lymphocytes, leading to lymphocyte activation, which is followed by the firm adhesion of lymphocytes on the HEVs using leukocyte function-associated antigen 1 (LFA-1) and intercellular adhesion molecule 1 (ICAM-1). In contrast, the adhesion molecules and chemokines that control the migration of lymphocytes from peripheral tissues into draining LNs are not well understood. Previous in vitro and in vivo studies suggest that Clever-1 and the mannose receptor on lymphatic endothelia mediate the binding of lymphocytes to lymphatic endothelia and/or for the migration of lymphocytes through the vessel wall [23,24,25,26]. It has also been shown that CCR7 is important for the migration of lymphocytes from normal and inflamed skin, inflamed lungs, and peritoneal cavity into draining LNs [27,28,29,30].

The binding of lymphocyte LFA-1 to ICAM-1 and/or ICAM-2 on blood vessel endothelia is important for the migration of lymphocytes into secondary lymphoid tissues and inflamed peripheral tissues [4,6,15,31,32,33,34,35]. The LFA-1and ICAM-1 adhesion pathway is also important for the migration of dendritic cells from peripheral tissues through lymphatic vessels into draining LNs [36,37]. Thus, we hypothesize that LFA-1 and ICAM-1 are important for the homeostatic migration of lymphocytes from peripheral tissues into draining LNs. To test this hypothesis, we use an adoptive lymphocyte transfer model in which donor lymphocytes are transferred into peripheral tissues and their migration into draining LNs is assessed. Using LFA-1-deficient (LFA-1^-/-^) donor lymphocytes and ICAM-1^-/-^ host mice, we find that the LFA-1/ICAM-1 adhesion pathway is important for the homeostatic migration of lymphocytes from peripheral tissues into LNs.

## 2. Materials and Methods

### 2.1. Mice

LFA-1^-/-^ (Itgal^tm1bII^/J), ICAM-1^-/-^ (Icam1^tm1bay^/J), and wild-type (WT) C57BL/6 (CD45.2) mouse breeding pairs were obtained from The Jackson Laboratory Bar Harbor, ME, USA. WT C57BL/6 (CD45.1) mice have been previously described [38]. These mice were bred and housed in the Stanford University Animal Facility. Female mice at 4–6 weeks of age were used in all experiments, unless indicated. Stanford University’s Administrative Panel of Laboratory Animal Care approved all the studies.

### 2.2. Antibodies and Other Reagents

Fluorochrome-conjugated monoclonal antibodies (mAbs) against B220 (RA3-6B2), CD3 (145-2C11), CD4 (GK 1.5), CD8 (53.6.7), CD11b (M1/70), CD44 (IM7), CD45.1 (A20), and CD45RB (C363.16A) were purchased from eBioscience (San Diego, CA, USA). Goat anti-lymphatic vessel endothelial receptor-1 (LYVE-1) polyclonal antibody (Ab) and rat anti-LFA-1 mAb (2D7) were purchased from R&D System (Minneapolis, MN, USA) and BD Biosciences (San Diego, CA, USA), respectively. An anti-ICAM-1 mAb-producing hybridoma (BE29) was obtained from the American Type Culture Collection (Manassas, VA, USA). Alexa Fluor 488-donkey anti-rat IgG Ab, Alexa Fluor 546-donkey anti-goat IgG Ab, and carboxyfluorescein succinimidyl ester (CFSE) were purchased from Molecular Probes (Eugene, OR, USA). Type 1A collagenase and DNase 1 were purchased from Sigma (St. Louis, MO, USA).

### 2.3. Tissue Immunofluorescence Staining of ICAM-1

Skin, lungs, and pancreas, and their draining LNs (cervical LN (CLN), bronchial LN (BLN), and pancreatic LN (PanLN), respectively) were harvested from 6-wk-old female WT mice, frozen, and sectioned. Acetone-fixed sections were incubated with Abs against LYVE-1 and ICAM-1 or with negative control Abs for 1 h and washed in phosphate-buffered saline (PBS). The sections were then incubated with Alexa Fluor 546-donkey anti-goat IgG Ab (to detect LYVE-1) and Alexa Fluor 488-donkey anti-rat IgG Ab (to detect ICAM-1) in PBS with 4% normal mouse serum for 45 min. After three washes with PBS, the sections were mounted, cover-slipped, and imaged on an Olympus fluorescence microscope (BX60, Olympus America Inc., Center Valley, PA, USA) equipped with a QImaging digital camera (Retiga 2000R, Surrey, British Columbia, Canada) using Image-Pro Plus software (Version 6.2.0.424, Media Cybernetics Inc., Bethesda, MD, USA).

### 2.4. Flow Cytometric Analysis of LFA-1 Expression on Peripheral Tissue Lymphocytes

Lymphocytes for flow cytometric analysis were isolated from the skin, lungs, and peritoneum of 6-wk-old female WT mice. Briefly, the mice were perfused with 50 mL heparin-PBS through the left and right ventricles, respectively, and the skin or lungs were excised. Visible lymphoid tissue was removed, and the remaining tissue was finely minced and incubated at 37 °C for 1 h in RPMI-1640 medium containing 0.5 mg/mL type 1A collagenase, 0.01% DNase 1, 5% FCS, and 10 mM HEPES. After washing extensively with RPMI-1640 medium and removing tissue particulates, lymphocytes were isolated using Lympholyte cell separation media (Cedarlane Lab Ltd., Burlington, ON, Canada). Peritoneal lymphocytes were obtained by lavaging the peritoneum with 10 mL of PBS. Lymphocyte suspensions were stained with FITC-anti CD11b (to exclude B1 B cells), PE-anti-LFA-1, PE-Cy7-anti-CD3, and Alexa Fluor 647-anti-B220 mAbs or with conjugation- and isotype-matched negative-control monoclonal antibodies (mAbs). The expressions of LFA-1 on T cells (CD3^+^ cells) and (conventional or B2) B cells (CD11b^-^B220^+^ cells) were determined by analyzing the cells in the lymphocyte gate on a FACSCalibur flow cytometer (BD Biosciences, San Diego, CA, USA) [15].

### 2.5. In Vivo Lymphocyte Migration Assays

Short-term in vivo lymphocyte migration assays were used to evaluate the ability of each subset of lymphocytes to migrate from peripheral tissues or blood vessels into LNs and the roles of LFA-1 and ICAM-1 in the migration of specific subsets of lymphocytes from peripheral tissues or blood vessels into LNs. Briefly, donor lymphocytes were isolated from the LNs and spleens of WT and/or LFA-1^-/-^ mice, and labeled with CFSE as previously described [15]. Flow cytometric analysis confirmed that all donor lymphocytes expressed LFA-1 at the levels indistinguishable from the lung, skin, and peritoneal cavity. To evaluate the role of ICAM-1, 5.0 × 10^7^ CFSE-labeled lymphocytes from WT CD45.2 or CD45.1 mice were transferred subcutaneously (s.c.) in the ear, intranasally (i.n.), intraperitoneally (i.p.), or intravenously (i.v.) into WT CD45.2 and ICAM-1^-/-^ CD45.2 host mice. To evaluate the role of LFA-1, 5.0 × 10^7^ CFSE-labeled WT CD45.1 lymphocytes combined with 5.0 × 10^7^ CFSE-labeled LFA-1^-/-^ CD45.2 lymphocytes were transferred s.c., i.n., i.p., or i.v. into WT CD45.2 host mice. In all experiments, host mice were sacrificed 16 h after transfer into tissues, or 2 h after i.v. transfer. Lymphocyte suspensions were prepared from the draining LNs (CLN, BLN, PanLN) and inguinal LNs (ILN) (as non-draining LNs), and stained with PE-anti-CD45.1, PE-Cy7-anti-CD3 and Alexa Fluor 647-anti-B220 mAbs. The percentages of donor T and B cells in host LNs and in the input population were determined by analyzing 2.0 × 10^5^ cells in the lymphocyte gate on an FACSCalibur flow cytometer.

In the experiments assessing the role of ICAM-1 in the migration of naive CD4^+^ T cells and memory CD4^+^ T cells from peripheral tissues into draining LNs, donor lymphocytes were isolated from the spleens of >1-year-old WT mice, labeled with CFSE, and injected s.c., i.n., or i.p. into WT mice and ICAM-1^-/-^ mice. Sixteen hours after the transfer, lymphocyte suspensions were prepared from the LNs of host mice, stained with Alexa Fluor 647-anti-CD4, PE-Cy7-anti-CD44, and PE-anti-CD45RB mAbs, and evaluated using flow cytometric analysis as described above.

The ability of donor lymphocytes to migrate from peripheral tissues into draining LNs was expressed as the ratio of donor cells in draining LNs to donor cells in non-draining LNs. The migration of LFA-1^-/-^ lymphocytes into LNs of WT mice was expressed as the ratio of LFA-1^-/-^ donor cells to WT donor cells in LNs of host mice. The migration of donor lymphocytes into LNs of ICAM-1^-/-^ mice was expressed as the percentage of the migration into LNs of WT host mice, in which the migration was set at 100%.

### 2.6. Analysis of Lymphocyte Subsets in LNs

Lymphocytes were isolated from the PLN (cervical, axillary, and inguinal LNs), BLN, and PanLN of 6-wk-old female WT, ICAM-1^-/-^, and LFA-1^-/-^ mice. The total cell number of each organ-draining LN was determined using a hemacytometer. To determine the percentage of each subset of lymphocytes, lymphocyte suspensions were stained with Alexa Fluor 405-anti-B220, APC-anti-CD3, APC-Alexa Fluor 750-anti-CD4, PE-Cy7-anti-CD8, PE-anti-CD44, and FITC-anti-CD45RB mAbs, and analyzed on a BD LSR II flow cytometry. The absolute number of cells in each subset was calculated by multiplying the total cell number by the percentage of each subset of lymphocytes.

### 2.7. Data Analysis

Student’s *t*-test was used to test the difference between two groups. *p* < 0.05 was considered to be statistically significant.

## 3. Results

### 3.1. ICAM-1 Is Expressed on Sinus Endothelia in LNs but Not on Lymphatic Vessel Endothelia in Peripheral Tissues

To evaluate the expression of ICAM-1 on lymphatic vessel endothelia in peripheral tissues and on sinus endothelia in LNs, we stained frozen sections of the skin, lungs, pancreas, and corresponding draining LNs of naive C57BL/6 mice with an anti-ICAM-1 mAb and an anti-LYVE-1 Ab (to stain endothelia of lymphatic vessels and LN sinuses). ICAM-1 was highly expressed on most endothelia of LYVE-1^+^ subcapsular sinuses (Figure 1A–C) and LYVE-1^-^ blood vessels in LNs (Figure 1D–F). ICAM-1 was also weakly expressed on the endothelia of LN cortical and medullary sinuses (Figure 1D–F). In contrast, ICAM-1 was not expressed on LYVE-1^+^ lymphatic vessel endothelia in the skin, lungs, and pancreas; although, it was expressed on LYVE-1^-^ blood vessel endothelia in these tissues (Figure 1G–I). Normal goat IgG (negative control for the anti-LYVE-1 Ab) and isotype negative-control mAb for ICAM-1 did not stain these tissues.

### 3.2. LFA-1 Is Expressed on Peripheral Tissue Lymphocytes

To evaluate the expression of LFA-1 on peripheral tissue lymphocytes, we isolated lymphocytes from the skin, lungs, and peritoneum of naive C57BL/6 mice and performed immunofluorescence staining and flow cytometric analysis (Figure 1J). LFA-1 was expressed on >95% of T cells from each tissue, with the highest expression on skin T cells, and on >70% of B cells, with a bimodal expression pattern on skin B cells. In each tissue, T cells expressed higher levels of LFA-1 than B cells.

### 3.3. T Cells Migrate More Efficiently Than B Cells from Peripheral Tissues into Draining LNs

We focused on the homeostatic migration of lymphocytes from the skin [27], the lungs [39], and the peritoneum [40] into their corresponding draining LNs. As our first step, we evaluated the ability of lymphocytes to migrate from peripheral tissues into draining and non-draining LNs. For this purpose, CFSE-labeled lymphocytes from LNs and spleens of 4–5-wk-old WT mice were s.c. (ear skin), i.n. (lung), or i.p. (peritoneum) transferred to age-matched WT host mice. Sixteen hours after cell transfer, more donor lymphocytes were found in the draining LNs as compared to non-draining LNs (ILNs) (Figure 2A). Donor T and B cells migrated into draining LNs at least 15-times more efficiently than into non-draining LNs after cell transfer (Figure 2B,C). Thus, adoptively transferred donor lymphocytes migrated preferentially into draining LNs, allowing us to assess their migration into draining LNs through the afferent lymph vessels.

Then, to evaluate the relative abilities of T and B cells to migrate from peripheral tissues into draining LNs, CFSE-labeled lymphocytes from WT mice were transferred s.c., i.n., or i.p. into age-matched WT host mice. As shown in Figure 3A, donor T cells migrated into draining LNs more efficiently than donor B cells. The ratios of donor T to B cells in the CLN, BLN, and PanLN of host mice were 20.8, 4.6, and 6.1, respectively, when the ratio of T to B cells in host LNs was normalized with the ratio of T to B cells in the input donor population. We also compared the abilities of T and B cells to migrate from blood vessels into the LNs of host mice 2 h after i.v. cell transfer. We found that T cells migrated more efficiently than B cells from blood vessels into CLN, BLN, and PanLN (Figure 3B).

Furthermore, to evaluate the relative abilities of naive CD4^+^ T cells and memory CD4^+^ T cells to migrate from peripheral tissues into draining LNs, CFSE-labeled lymphocytes from >1-year-old WT mice were transferred s.c., i.n., or i.p. into young WT host mice. Naive CD4^+^ T cells migrated from peripheral tissues into the draining LNs of host mice slightly more efficiently than memory CD4^+^ T cells (Figure 3C). As expected, naive CD4^+^ T cells migrated from the blood vessels more efficiently than memory CD4^+^ T cells (Figure 3D).

### 3.4. ICAM-1 Deficiency in Host Tissues Impairs the Migration of Lymphocytes from Peripheral Tissues into Draining LNs, but Not from Blood Vessels into LNs

To investigate the role of ICAM-1 in lymphocyte migration from peripheral tissues into draining LNs, we transferred WT donor lymphocytes into the peripheral tissues of WT and ICAM-1^-/-^ mice. As shown in Figure 4, WT donor T and B cells migrated from the skin and lungs into the draining LNs in ICAM-1^-/-^ mice < 30% as efficiently as in WT mice (Figure 4A,B,D). Similarly, WT T and B cells migrated from the peritoneum into the PanLN of ICAM-1^-/-^ mice at approximately 30% and 50%, respectively, as efficiently as in WT mice (Figure 4A,B,D). In contrast, donor T cells migrated from blood vessel HEVs into LNs in ICAM-1^-/-^ host mice equally well, or even slightly better than they did in WT host mice 2 h after i.v. transfer (Figure 4C). There was a slight reduction in the migration of donor B cells from blood vessel HEVs into CLN, but not the BLN and PanLN of ICAM-1^-/-^ host mice as compared to WT host mice (Figure 4E). These results indicate that ICAM-1 is important for the migration of lymphocytes from peripheral tissues into LNs, but not for the migration of most lymphocytes from blood vessels into LNs.

### 3.5. LFA-1-Deficient Lymphocytes Migrate Poorly from Peripheral Tissues into Draining LNs

We evaluated the abilities of CFSE-labeled WT lymphocytes (CD45.1) and CFSE-labeled LFA-1^-/-^ lymphocytes (CD45.2) to migrate from peripheral tissues into the draining LNs of WT mice (CD45.2). LFA-1^-/-^ donor T cells migrated 10–20% as efficiently as WT donor T cells, into the CLN, BLN, and PanLN (Figure 5A,B). Similarly, LFA-1^-/-^ donor B cells migrated 30–40% as efficiently as WT donor B cells into these LNs (Figure 5A,D). We also evaluated the migration of LFA-1^-/-^ donor lymphocytes from the blood vessels into LNs. LFA-1^-/-^ donor T cells migrated from blood vessels into the CLN and BLN 10% as efficiently, and into PanLN 40% as efficiently as WT donor T cells (Figure 5C). Similarly, LFA-1^-/-^ donor B cells migrated from blood vessels into CLN and BLN 30% as efficiently, and into PanLN 70% as efficiently as WT donor B cells (Figure 5E). These results indicate that lymphocyte LFA is important for the migration of lymphocytes from both peripheral tissues and blood vessels into LNs.

### 3.6. Lack of the LFA-1/ICAM-1 Adhesion Pathway Reduces the Number of Memory T Cells in LNs

Having demonstrated the importance of LFA-1 and ICAM-1 in the migration of lymphocytes from peripheral tissues into draining LNs, we investigated whether the LFA-1/ICAM-1 pathway contributed to the homeostatic maintenance of specific subsets of lymphocytes in LNs. For this purpose, we isolated lymphocytes from the PLNs (cervical, axillary, and inguinal LNs), BLNs, and PanLNs of WT, ICAM-1^-/-^, and LFA-1^-/-^ mice, and determined the absolute numbers of subsets of lymphocytes.

There were significantly fewer B cells (B220^+^ cells) and naive CD4^+^ T cells (CD3^+^CD4^+^ CD44^low-high^CD45^high^ cells) in the PLNs of ICAM-1^-/-^ mice as compared to WT mice (Figure 6A,B). As expected, there were significantly fewer B cells, naive CD4^+^ T cells, and naive CD8^+^ T cells (CD3^+^CD8^+^CD44^low-med^cells) in the PLNs and BLNs of LFA-1^-/-^ mice as compared to WT mice (Figure 6A–C). The deficiency of either ICAM-1 or LFA-1 did not significantly affect the absolute numbers of B cells, naïve CD4^+^ T cells, or naive CD8^+^ T cells in the PanLNs (Figure 6A–C). However, the absolute numbers of memory CD4^+^ T cells (CD4^+^CD44^high^CD45RB^low^) and memory CD8^+^ T cells (CD3^+^CD8^+^CD44^high^) in the LNs of ICAM-1^-/-^ and LFA-1^-/-^ mice were significantly reduced as compared to WT mice (Figure 6D,E). Thus, the LFA-1/ICAM-1 pathway is important in maintaining memory T cellularity in LNs.

### 3.7. ICAM-1 Deficiency in Host Tissues Impairs the Migration of Both Naive and Memory CD4^+^ T Cells from Peripheral Tissues into Draining LNs

To determine whether the reduction in the number of memory T cells in the LNs of ICAM-1^-/-^ mice resulted from a selective influence of ICAM-1 deficiency on the migration of memory T cells, as compared to naive T cells, from the peripheral tissues into draining LNs, we transferred donor lymphocytes from >1-year-old WT mice into the peripheral tissues of young WT and ICAM-1^-/-^ mice. Sixteen hours after the transfer, we assessed the relative number of donor naive CD4^+^ T cells and memory CD4^+^ T cells in draining LNs of the host mice. As shown in Figure 7, both subsets of CD4^+^ T cells migrated 70–80% less efficiently from the skin and lungs into CLNs and BLNs, and >90% less efficiently from peritoneum into the PanLNs in ICAM-1^-/-^ host mice than they did in WT mice. Thus, ICAM-1 deficiency in mice substantially inhibited the migration of naive CD4^+^ T cells and memory CD4^+^ T cells from peripheral tissues into draining LNs.

## 4. Discussion

ICAM-1 on endothelia is important for the migration of lymphocytes from blood vessels into secondary lymphoid tissues and inflamed peripheral tissues [4,6,41,42]. In this study, we demonstrated that ICAM-1 was highly expressed on LYVE-1^+^ endothelia in the subcapsular sinuses of LNs (Figure 1). Previous studies reported that ICAM-1 was expressed on mouse lymphatic endothelial cell lines, human primary dermal lymphatic endothelia, and lymphatic vessels in the human tongue [43,44,45]. Our tissue immunofluorescence staining did not detect the ICAM-1 on LYVE-1^+^ lymphatic vessels of the non-inflamed normal skin, lungs, and pancreas of naive mice. Our findings are consistent with a previous study showing the expression of ICAM-1 on the lymphatic vessels of inflamed, but not uninflamed, skin in mice [44]. We also found that LFA-1 was expressed on most T and B cells in peripheral tissues, such as the skin, lungs, and peritoneum. Thus, ICAM-1 on the sinus endothelia of LNs is available for binding LFA-1^+^ lymphocytes emerging from the afferent lymph vessel.

Donor lymphocytes migrated significantly less well from peripheral tissues into draining LNs in ICAM-1^-/-^ mice than in WT mice (Figure 4A,B,D). In contrast, donor lymphocytes migrated equally well from blood vessels into the LNs of ICAM-1^-/-^ and WT mice (Figure 4C,E). Several studies have addressed the role of ICAM-1 in the migration of lymphocytes from blood vessels into LNs and peripheral tissues. In in vitro transendothelial migration studies [46,47], WT T cells migrated poorly through ICAM-1^-/-^ or ICAM-1/ICAM-2 double-deficient endothelia; however, impaired migration was restored when the endothelia were transfected with ICAM-1, ICAM-2, or both. In short-term in vivo lymphocyte migration assays, WT lymphocytes migrated poorly from blood vessels into the LNs of ICAM-1/ICAM-2 double-deficient mice, as compared to ICAM-1^-/-^ and WT mice, indicating that ICAM-1’s role in this migration was replaced by ICAM-2 [34]. In an intravital microscopy study, however, the deficiency of ICAM-1, but not ICAM-2, significantly impaired lymphocyte arrest on PLN HEVs and lymphocyte motility in LN parenchyma [48]. In our study, the migration of donor T cells from peripheral tissues into the draining LNs of ICAM-1^-/-^ host mice was 70% less efficient as compared to WT host mice (Figure 4B). Thus, ICAM-1 is the major ligand for the LFA-1 in migration of lymphocytes from peripheral tissues into draining LNs.

In our in vivo lymphocyte migration assays, T and B cells from LFA-1^-/-^ mice migrated poorly from peripheral tissues into draining LNs (Figure 5). It was well established that LFA-1 was important for the migration of lymphocytes from blood vessels into secondary lymphoid tissues and inflamed peripheral tissues and for the migration of dendritic cells from peripheral tissues into draining LNs [15,31,32,36]. Our findings indicate that LFA-1 is also required for the homeostatic migration of lymphocytes from peripheral tissues into LNs.

Either LFA-1 or ICAM-1 deficiencies reduced memory T cells in the mouse LNs (Figure 6D,E). This is a previously unrecognized phenotype in either ICAM-1^-/-^ or LFA-1^-/-^ mice [31,49,50,51]. Several pieces of our data suggest that this phenotype might result, at least in part, from the impaired entry of memory T cells from peripheral tissues into LNs in these mice. First, donor memory CD4^+^ T cells migrated more efficiently from peripheral tissues into LNs than from blood vessel HEVs (Figure 3C). Second, the absence of ICAM-1 in host mice significantly impaired the migration of WT donor T cells from peripheral tissues into draining LNs (Figure 5A,B), but not the migration of T cells from blood vessels into LNs (Figure 5C). Third, the relative numbers of memory CD4^+^ and CD8^+^ T cells were significantly lower in the LNs of ICAM-1^-/-^ mice and LFA-1^-/-^ mice as compared to WT mice. Fourth, given that memory CD4^+^ T cells are overrepresented in the afferent lymph vessel, the migration of memory CD4^+^ T cells from peripheral tissues to LNs will be much more dramatically affected by ICAM-1 deficiency. Thus, the LFA-1/ICAM-1 adhesion pathway in part contributed to the maintenance of the memory T cellularity in LNs.

The absence of the LFA-1 or ICAM-1 reduced the migration of donor lymphocytes from peripheral tissues into draining LNs by 70–90%. We found that ICAM-2, another ligand for LFA-1, was also expressed on LYVE-1^+^ sinus endothelia in LNs, but not on the lymphatic vessels in peripheral tissues. Additionally, chloride channel calcium-activated 1, an additional ligand for LFA-1, which is expressed on lymphatic vessels but not blood vessels in peripheral tissues and on sinuses in LNs was observed [52,53]. Thus, ICAM-2 and chloride channel calcium-activated 1 may play a minor role in lymphocyte migration from peripheral tissues into LNs. Previous studies have demonstrated the roles of several molecules in regulating the migration of lymphocytes from peripheral tissues into draining LNs. For example, lymphocyte CCR7 promoted the migration of lymphocytes from the peritoneal cavity, inflamed lung, and normal and inflamed skin into draining LNs [27,28,29,30]. The deficiency of the mannose receptor on lymphatic vessels in peripheral tissues and sinuses in LNs impaired the migration of lymphocytes from the skin into draining LNs [54]. Anti-CLEVER-1 antibody blocked the migration of adoptive transferred donor lymphocytes from footpad into LNs in rabbits [26]. Sphingosine 1-phosphate receptor 1 that regulates lymphocyte egress from LNs prevented T-cell migration from the skin into draining LNs [55,56,57]. In in vitro translymphatic endothelial migration assays, blocking LFA-1 and ICAM-1 inhibited the transmigration of T cells and dendritic cells [44,55]. Taken together, our study extends the functions of the LFA-1/ICAM-1 adhesion pathway, indicating its novel role in controlling lymphocyte migration from peripheral tissues into LNs.

Similar to previous in vivo studies [27,28,54,58], we were unable to determine the sites (peripheral tissues and/or LNs) and mechanisms (adhesion and/or transmigration) by which the LFA-1/ICAM-1 pathway controls the migration of lymphocytes from peripheral tissues into draining LNs. It has been recently shown that initial lymphatic vessels in peripheral tissues have gaps for the intravasation of dendritic cells into lymphatic lumen [59]; thus, a similar mechanism may also be used by lymphocytes in peripheral tissues to gain access to the lymphatic lumen in an ICAM-1-independent manner. This is supported by our findings that ICAM-1 is expressed on sinus endothelia in LNs, but not on lymphatic vessels in peripheral tissues. Thus, the recognition of lymphocyte LFA-1 by ICAM-1 on LN sinus endothelia enables the extravasation of lymphocytes from afferent lymphatic vessels into LNs.

Other mechanisms were also reported. For example, intravital imaging studies revealed that ICAM-1 and LFA-1 mediated T cells crawling within inflamed lymphatic capillaries and subsequently entered draining LNs [60]. Neutrophils emigrated from inflamed skin to draining LNs using LFA-1, macrophage 1 antigen, C-X-C-motif chemokine receptor 4, and sphinosine-1-phosphate receptor [61]. Regulatory T cells upregulated the expression of vascular endothelial adhesion molecule 1 and C-C motif chemokine ligand 21 on lymphatic endothelial cells thus enhancing T-cell migration from peripheral tissues to draining LNs via lymphotoxin αβ/lymphotoxin β receptor signaling in both steady and inflamed states [62]. Additionally, T-cell glycogen synthase kinase β3 also regulated LFA-1/ICAM-1-mediated T-cell migration in vitro by interacting with cytoskeletal regulator collapsing response mediator protein 2 and neurogenic locus notch homolog protein 1 [63].

Additionally, two issues need to be addressed regarding the present study. First, lymphocyte migration from blood vessels into secondary lymphoid and peripheral tissues is a multistep cascade that involves adhesion molecules on both lymphocytes and vascular endothelial cell tissue [15,20,21,22]. For example, peripheral node addressin/L-selectin, mucosal adhesion molecule-1/α4β7 integrin, and E-selectin/its lymphocyte receptor mediate circulating lymphocyte migration to peripheral LNs and bronchus-associated lymph tissues, Payer’s patches, and the skin, respectively [6,15,22]. Lymphocyte LFA-1/vascular endothelial ICAM-1 is the shared pathway for the migration of circulating lymphocytes by causing the firm adhesion and sticking of lymphocytes to endothelial cells and subsequent diapedesis into tissues [6]. Though it is not known whether multistep cascades exist for lymphocyte migration from lymphatic vessels into draining LNs, integrins, including α4, β1, β7, and αV, indeed played a partial role in mediating lymphocyte migration to LNs via lymphatic vessels in certain circumstances [64,65]. Almost all lymphocytes express α4, β1, and β7 integrins, and certain subsets of lymphocytes express αV integrin [22,66,67]. The presence of corresponding ligands, including vascular cell adhesion molecule-1 (for α4β1), mucosal cell addressin-1 (for α4β7 integrin), and fibronectin (for αvβ1 integrin), on lymphatic endothelial cells may determine whether these integrins contribute to lymphocyte migration to LNs via lymphatic vessels. However, vascular cell adhesion molecule-1 and mucosal cell addressin-1 were expressed on lymphatic vessels under inflamed conditions or lymphatic endothelial cells after exposure to inflammatory mediators, but not on lymphatic vessels in uninflamed tissues or a steady state [43,44,55,62,68,69,70,71,72,73,74,75]. Because the present study focused on the LFA-1/ICAM-1 pathway, we did not assess the roles of α4, αV, β1, β7, and integrins, and their lymphatic endothelial ligands in lymphocyte migration from lymphatic vessels to draining LNs. Thus, we cannot rule out that these lymphocyte integrins may be potentially responsible for the small residual migration of adoptively transferred lymphocytes from lymphatic vessels to draining LNs in the absence of the LFA-1/ICAM-1 pathway. Second, all donor lymphocytes isolated from WT mice used for in vivo migration assays expressed LFA-1. Almost all lymphocytes from WT mouse peripheral tissues, at least the skin, lungs, and peritoneal cavity, also expressed LFA-1 although LFA-1 expression levels slightly differed among tissues. The endpoint of our study was to analyze the subsets of donor lymphocytes recovered from host lymph nodes, regardless of whether donor lymphocytes were transferred via intravenous, intraperitoneal, subcutaneous injections, or intranasal installation. Thus, we neither assessed the correlation of input donor lymphocyte LFA-1 levels with those of donor cells harvested from host LNs nor the difference in LFA-1 levels in LNs between WT and ICAM-1^-/-^ host mice.

In conclusion, we showed that LYVE-1^+^ sinus endothelia of LNs, but not lymphatic vessels of peripheral tissues, expressed ICAM-1. In vivo lymphocyte migration assays demonstrated that the LFA-1/ICAM-1 adhesion pathway was important for the migration of lymphocytes from peripheral tissues to LNs and contributed to the homeostatic maintenance of lymphocyte populations, particularly memory T cells in LNs.

## Figures and Tables

**Figure 1 biomolecules-13-01194-f001:**
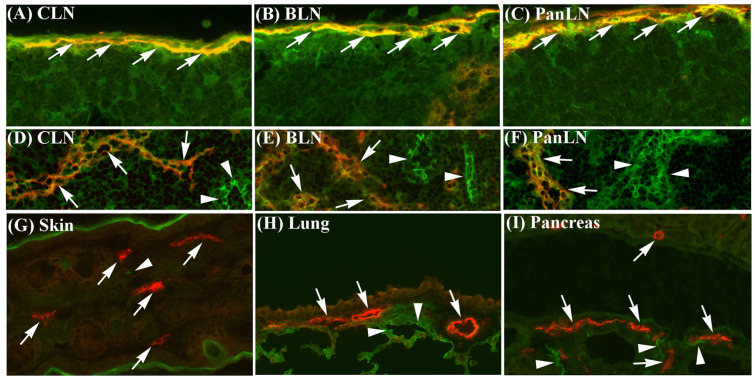
Intercellular adhesion molecule-1 (ICAM-1) is expressed on sinus endothelia in lymph nodes (LNs) but not on lymphatic vessel endothelia in peripheral tissues, and leukocyte function-associated antigen-1 is expressed on lymphocytes in peripheral tissues. (**A**–**I**): Frozen sections of the skin, lungs, and pancreas, and their draining LNs from 6-wk-old wild-type (WT) C57BL/6 mice were stained with antibodies against lymphatic vessel endothelial receptor-1 (LYVE-1) (red identifies lymphatic and sinus endothelia, arrows) and ICAM-1 (green), or with negative-control antibodies. (**A**–**C**): ICAM-1 was highly expressed on LYVE-1^+^ subcapsular sinus endothelia (yellow, arrows) in cervical LN (CLN, which drains the ear skin), bronchial LN (BLN, which drains the lungs), and pancreatic LN (PanLN, which drains the pancreas and peritoneum). (**D**–**F**): ICAM-1 was highly expressed on LYVE-1^-^ blood vessel endothelia (green, arrowheads) and weakly expressed on endothelia of cortical sinuses (yellow, arrows) in CLN, BLN, and PanLN. (**G**–**I**): ICAM-1 was highly expressed on LYVE-1^-^ blood vessel endothelia (green, arrowheads), but not on LYVE-1^+^ lymphatic vessel endothelia (red, arrows) in the skin, lungs, and pancreas. Neither goat IgG (negative control for anti-LYVE-1 antibody) nor the isotype control mAb for ICAM-1 stained these tissues. (frozen-section immunofluorescence stains; original magnification = 200×). (**J**): Lymphocytes were isolated from the skin, lungs, and peritoneum of 6-wk-old WT C57BL/6 mice, stained with monoclonal antibodies (mAbs) against CD3 (T cells), B220 (B cells), CD11b, and leukocyte function-associated antigen-1 (LFA-1), or with mAbs against CD3, B220, CD11b, and an isotype control mAb (for LFA-1), and evaluated by flow cytometric analysis. Overlay histogram in each panel shows staining with an anti-LFA-1 mAb (unshaded) and isotype control mAb (shaded) on T cells (**upper** panels) or conventional (or B2) B cells (B220^+^CD11b^-^) (**lower** panels). The numbers in each panel indicate the percentage of cells that are LFA-1^+^ (**top**) and mean fluorescence intensity of LFA-1 staining (**bottom**).

**Figure 2 biomolecules-13-01194-f002:**
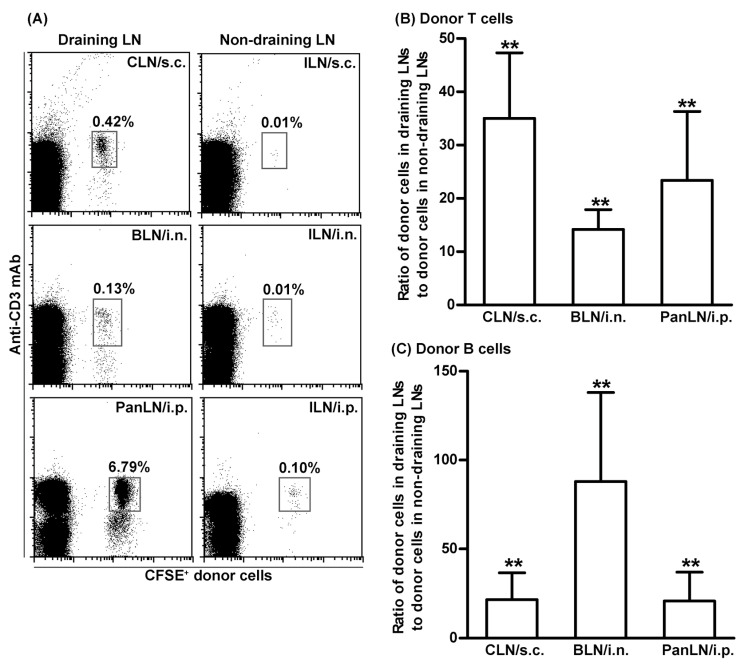
Lymphocytes migrate efficiently from peripheral tissues into draining lymph nodes. Fifty million carboxyfluorescein succinimidyl ester (CFSE)-labeled lymphocytes from young wild-type (WT) mice were transferred subcutaneously (s.c.), intranasally (i.n.), or intraperitoneally (i.p.) into young WT host mice. Sixteen hours after transferring into tissues, donor lymphocytes in the draining LNs ((cervical LNs (CLNs) for s.c., bronchial LNs (BLNs) for i.n., and pancreatic LNs (PanLNs) for i.p., and non-draining inguinal LNs (ILNs)) were identified by immunofluorescence staining with monoclonal antibodies against CD3 and B220 and evaluated by flow cytometric analysis. (**A**): representative flow cytometric plots show donor T cells (CFSE^+^CD3^+^ cells) as the percentage of total lymphocytes in organ-draining LNs (left column) and non-draining LNs (right column) of host mice. (**B**,**C**): Donor T (**B**) and B (**C**) cells migrated into draining LNs at least 15-times more than into non-draining LNs after transferring into tissues. Student’s *t*-test, ** *p* < 0.01 compared with the theoretical input ratio, which is set as 1. *n* = 7–8 mice in each group.

**Figure 3 biomolecules-13-01194-f003:**
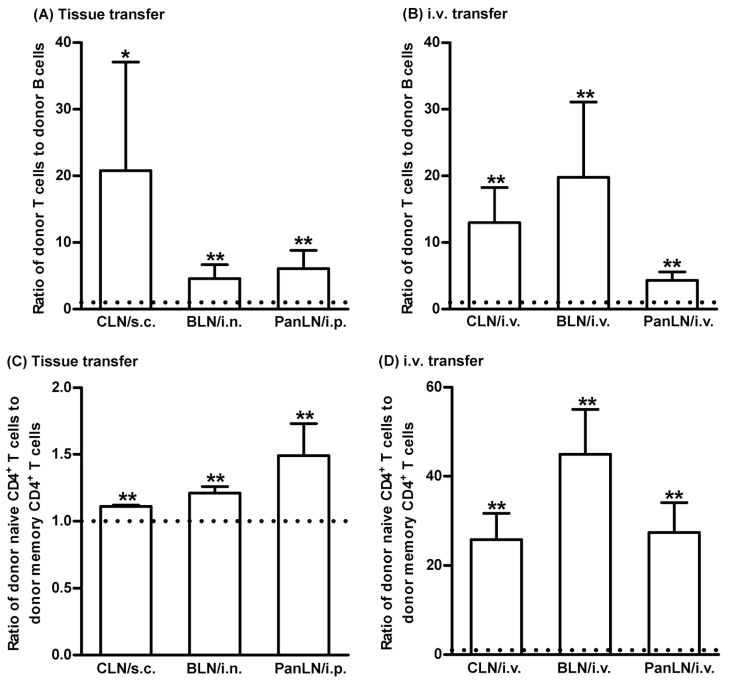
T cells migrate efficiently from peripheral tissues into draining lymph nodes. Fifty-million carboxyfluorescein succinimidyl ester-labeled lymphocytes from young (**A**,**B**) and old (**C**,**D**) wild-type (WT) mice were transferred subcutaneously (s.c.), intranasally (i.n.), intraperitoneally (i.p.), or intravenously (i.v.) into young WT mice. Sixteen hours after cell transfer into tissues (**A**,**C**) or 2 h after i.v. cell transfer (**B**,**D**), lymphocyte suspension of host cervical lymph nodes (CLNs), bronchial LNs (BLNs), or pancreatic LNs (PanLNs) were stained with monoclonal antibodies against CD3 and B220 (**A**,**B**) or against CD4, CD44, and CD45RB (**C**,**D**), and evaluated by flow cytometric analysis. The results are expressed as the ratio of donor T to B (**A**,**B**) cells or donor naive (CD44^low-high^CD45RB^high^) CD4^+^ T cells to donor memory (CD44^high^CD45RB^low^) CD4^+^ T cells (**C**,**D**) in each draining LN. The horizontal dotted line in each panel indicates the input ratio of the two subsets of lymphocytes, which is set as 1. Student’s *t*-test, * *p* < 0.05 and ** *p* < 0.01 compared to input ratio, *n* = 5–17 (**A**,**B**) and *n* = 4–5 (**C**,**D**), mice in each group.

**Figure 4 biomolecules-13-01194-f004:**
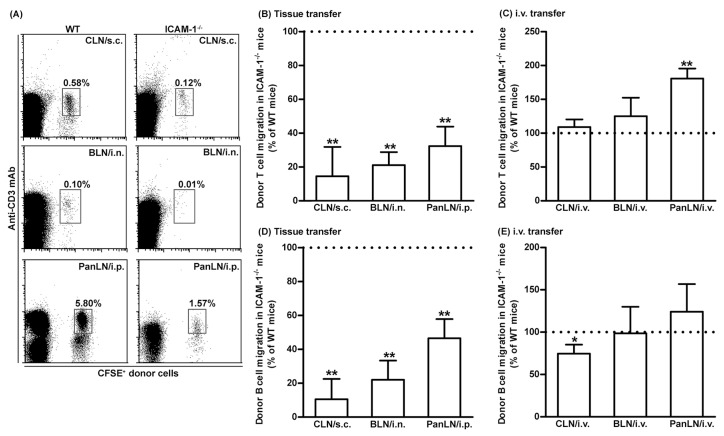
Intercellular adhesion molecules-1 (ICAM-1) is important for the migration of T and B cells from peripheral tissues into draining lymph nodes (LNs). Fifty-million carboxyfluorescein succinimidyl ester (CFSE)-labeled wild-type (WT) lymphocytes were transferred into the tissues or blood vessels of WT and ICAM-1-deficient (ICAM-1^-/-^) mice. Sixteen hours after cell transfer into tissues or 2 h after intravenous (i.v.) transfer, donor lymphocytes in the LNs of host mice were identified by immunofluorescence staining with monoclonal antibodies against CD3 and B220 and evaluated by flow cytometric analysis. (**A**): Representative flow cytometric plots show the percentage of donor T cells (CFSE^+^CD3^+^ cells) in total lymphocytes in organ-draining LNs of WT (**left** column) and ICAM-1^-/-^ (**right** column) host mice after cell transfer into each tissue. (**B**–**E**): Donor T (**B**) and B (**D**) cells migrated less avidly into draining LNs of ICAM-1^-/-^ mice than of WT mice after cell transfer into each tissue. In contrast, donor T (**C**) and B cells migrated into the LNs in ICAM-1^-/-^ mice as well or better than into WT mice after i.v. cell transfer, except for B-cell migration into CLNs. The migration of donor T (**B**,**C**) and B (**D**,**E**) cells into the LNs of ICAM-1^-/-^ mice is presented as the percentage of the migration into the LNs of WT mice, where migration is set at 100%. Student’s *t*-test, * *p* < 0.5 and ** *p* < 0.01 compared to WT mice, respectively, *n* = 4–6 mice in each group.

**Figure 5 biomolecules-13-01194-f005:**
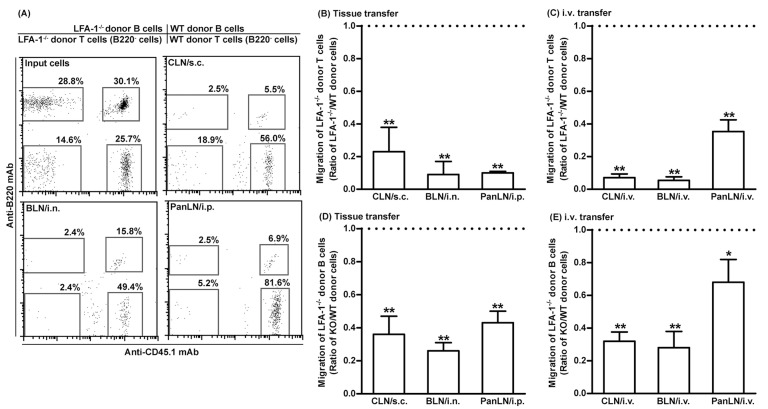
Leukocyte function-associated antigen-1 (LFA-1) is important for the migration of lymphocytes from peripheral tissues into draining lymph nodes (LNs). Fifty-million carboxyfluorescein succinimidyl ester-labeled wild-type (WT) CD45.1 lymphocytes and an equal number of carboxyfluorescein succinimidyl ester-labeled LFA-1-deficient (LFA-1^-/-^) CD45.2 lymphocytes were transferred into the tissues or blood vessels of WT CD45.2 mice. Sixteen hours after transfer into tissues or 2 h after intravenous (i.v.) transfer, donor T and B cells in host-mice LNs were evaluated by immunofluorescence staining with monoclonal antibodies against CD3, B220, and CD45.1, and evaluated by flow cytometric analysis. LFA-1^-/-^ donor T cells (**A**–**C**) and LFA-1^-/-^ B cells (**A**,**D**,**E**) migrated less avidly than WT donor cells into host-mouse LNs after transfer into tissues (**A**,**B**,**D**) or i.v. transfer (**A**,**C**,**E**). The migration of donor cells is expressed as the ratio of LFA-1^-/-^ donor T cells to WT donor T cells (**A**–**C**) or the ratio of LFA-1^-/-^ donor B cells to WT donor B cells (**A**,**D**,**E**). The horizontal dotted line in each panel indicates the input ratio of LFA-1^-/-^ donor T cells to WT donor T cells (**B**,**C**) or the input ratio of LFA-1^-/-^ donor B cells to WT donor B cells (**D**,**E**), which is set as 1. Student’s *t*-test, * *p* < 0.05 and ** *p* < 0.01 compared to input ratio, *n* = 4–5 mice in each group.

**Figure 6 biomolecules-13-01194-f006:**
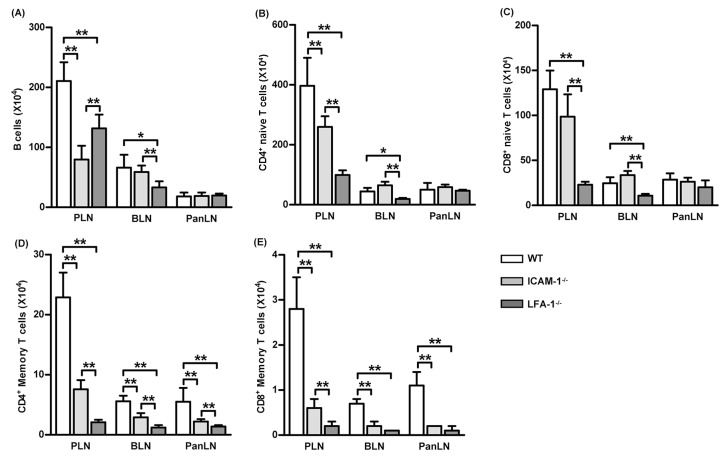
Memory T cells are reduced in the lymph nodes of intercellular adhesion molecule-1 (ICAM-1)-deficient and leukocyte function-associated antigen-1 (LFA-1)-deficient mice. Lymphocytes from the peripheral lymph nodes (PLNs), bronchial LNs (BLNs), and pancreatic LNs (PanLNs) of 6-wk-old female wild-type (WT, open bar), ICAM-1^-/-^ (light-gray bar), and LFA-1^-/-^ (dark-gray bar) mice were enumerated using a hemacytometer, then stained with fluorochrome-conjugated mAbs against subsets of lymphocytes and evaluated by FACS analysis. Data are the absolute numbers of B cells (B220^+^ cells) (**A**), naive CD4^+^ T cells (CD3^+^CD4^+^ CD44^low-high^CD45^high^ cells) (**B**), naive CD8^+^ T cells (CD3^+^CD8^+^CD44^low-med^cells) (**C**), memory CD4^+^ T cells (CD4^+^CD44^high^CD45RB^low^) (**D**), and memory CD8^+^ T cells (CD3^+^CD8^+^CD44^high^) (**E**). Student’s *t*-test, * *p* < 0.05, ** *p* < 0.01 compared to WT mice, *n* = 5 mice in each group.

**Figure 7 biomolecules-13-01194-f007:**
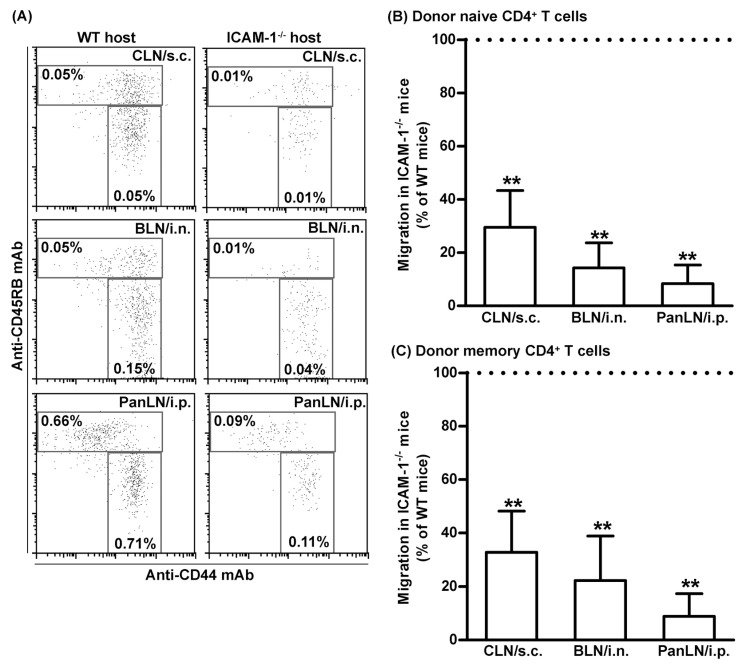
Intercellular adhesion molecule-1 (ICAM-1) deficiency in mice has an equal effect on the migration of naive CD4^+^ and memory CD4^+^ T cells from peripheral tissues into draining LNs. Fifty-million carboxyfluorescein succinimidyl ester-labeled lymphocytes from >1 year old wild-type (WT) mice were transferred into the tissues of WT and ICAM-1-deficient (ICAM-1^-/-^) mice. Sixteen hours after cell transfer, donor naive CD4^+^ T cells (**A**,**B**) and donor memory CD4^+^ T cells (**A**,**C**) in the cervical lymph nodes (CLNs), bronchial LNs (BLNs), and pancreatic LNs (PanLNs) of host mice were identified by immunofluorescence staining with monoclonal antibodies against CD4, CD44, and CD45RB, and evaluated by flow cytometric analysis. (**A**): Representative flow cytometric plots show the percentages of donor naive CD4^+^ (CD44^low-high^CD45RB^high^) and memory CD4^+^ (CD44^high^CD45RB^low^) T cells in the total lymphocytes of organ-draining LNs of WT (left column) and ICAM-1^-/-^ (right column) host mice after cell transfer into tissue. (**B**,**C**): Donor naive CD4^+^ (**B**) and memory CD4^+^ (**C**) T cells migrated less avidly from the peripheral tissue into draining LNs in ICAM-1^-/-^ host mice than in WT host mice. The migration of donor naive CD4^+^ (**B**) and memory CD4^+^ (**C**) T cells into the LNs of ICAM-1^-/-^ mice is expressed as the percentage of the migration into LNs of WT mice, which is set at 100%. Student’s *t*-test, ** *p* < 0.01 compared to WT mice, *n* = 4–5 mice in each group.

## Data Availability

All data were included within this article.

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
