# Peer review of "LFA-1/ICAM-1 Adhesion Pathway Mediates the Homeostatic Migration of Lymphocytes from Peripheral Tissues into Lymph Nodes through Lymphatic Vessels"

_biomolecules, 2023, doi:10.3390/biom13081194_

Round 1
Reviewer 1 Report
The authors describe here the migration of lymphocytes (B cells and T cells) from peripheral tissues to the lymph nodes mediated by LFA-1 and ICAM-1 adhesion. Using LFA-1 KO or ICAM-1 KO, the authors demonstrated that homing of B cells and T cells to cervical lymph nodes, bronchial lymph nodes and pancreatic lymph nodes was significantly reduced when LFA-1 or ICAM-1 was absent. While the role of LFA-1 and ICAM-1 in the migration of lymphocytes is clearly demonstrated in the current manuscript, major considerations involving interactions other than LFA-1/ICAM-1 are missing.
Comments:
1. It has been shown that the homing of lymphocytes involves not only LFA-1/ICAM-1 but also αV, β1 and β7 integrins (Martens et al Nature Communication 2020), and both LFA-1 and VLA-4 are required for the homing of T cells in the spleen (Chauveau et al Immunity 2020). Given that LFA-1 KO or ICAM-1 KO did not completely abolish the migration of T cells and B cells with 10–20% residual migration, the data suggest that interactions other than LFA-1/ICAM-1 are involved in mediating the migration of lymphocytes from peripheral tissues to the lymph nodes during homoeostasis. Even though the current study emphasises the LFA-1/ICAM-1 interaction, the authors should check the expression of integrins other than the β2 integrin in lymphocytes (Figure 1). The relative expression of the other integrin by lymphocytes may correlate to the residual migration activity observed and provide an explanation for how lymphocytes may sustain migration when LFA-1 or ICAM-1 is absent. The authors should consider using blocking antibodies against α4 or β7 integrin in LFA-1 KO or ICAM-1 KO mice to test whether the residual migration would be completely abolished.
2. Are the actual figures for Figure 4 and Figure 5 swapped? The graphs and data did not match the descriptions in the results or the figure legends.
3. Figure 1J describes T cells on the "left column" and B cells on the right column. Do you mean T cells on the top panel and B cells on the bottom panel?
4. Discussion should include what other interactions are known to mediate the homing of lymphocytes including α4 and β7 integrins which could account for the residual activity observed.
Reviewer 2 Report
This is an elegant manuscript presented by Guo et al showing the role of the LFA/ICAM pathway mediating migration of lymphocytes from the periphery to the LNs.

Round 2
Reviewer 1 Report
The revised manuscript is better than the original version.